# Pose ResNet: 3D Human Pose Estimation Based on Self-Supervision

**DOI:** 10.3390/s23063057

**Published:** 2023-03-12

**Authors:** Wenxia Bao, Zhongyu Ma, Dong Liang, Xianjun Yang, Tao Niu

**Affiliations:** 1School of Electronics and Information Engineering, Anhui University, Hefei 230601, China; 2Hefei Institutes of Physical Science, Chinese Academy of Sciences, Hefei 230031, China

**Keywords:** 3D human pose estimation, epipolar geometry, self-supervised learning, transfer learning, synthetic occlusion

## Abstract

The accurate estimation of a 3D human pose is of great importance in many fields, such as human–computer interaction, motion recognition and automatic driving. In view of the difficulty of obtaining 3D ground truth labels for a dataset of 3D pose estimation techniques, we take 2D images as the research object in this paper, and propose a self-supervised 3D pose estimation model called Pose ResNet. ResNet50 is used as the basic network for extract features. First, a convolutional block attention module (CBAM) was introduced to refine selection of significant pixels. Then, a waterfall atrous spatial pooling (WASP) module is used to capture multi-scale contextual information from the extracted features to increase the receptive field. Finally, the features are input into a deconvolution network to acquire the volume heat map, which is later processed by a soft argmax function to obtain the coordinates of the joints. In addition to the two learning strategies of transfer learning and synthetic occlusion, a self-supervised training method is also used in this model, in which the 3D labels are constructed by the epipolar geometry transformation to supervise the training of the network. Without the need for 3D ground truths for the dataset, accurate estimation of the 3D human pose can be realized from a single 2D image. The results show that the mean per joint position error (MPJPE) is 74.6 mm without the need for 3D ground truth labels. Compared with other approaches, the proposed method achieves better results.

## 1. Introduction

It has always been the goal of human society to allow machines to perceive and obtain information from the outside world through vision like human do. With the rapid development of human–computer interaction and computer vision technology, the issue of how to make machines understand and analyze human actions and behaviors correctly has become important. In this context, the task of human pose estimation was developed [1]. The 3D estimation of a human pose involves estimating the 3D positions of human joints from input images or videos. Human pose estimation is widely used in many fields, such as sports science, rehabilitation training, video surveillance and behavioral recognition [2,3].

In practical applications based on 3D human pose estimation, human motion information and joint positions can be obtained by wearing sensors and other devices, and the joint positions can be calculated by collecting depth maps through Microsoft Kinect [4]. However, the inconvenience of this operation and the high cost of the devices limit its range of applications. In contrast, ordinary cameras are common in daily life, and are used in monitoring equipment, digital photography and mobile phones. Hence, the estimation of a human pose based on ordinary RGB images has great research significance and application value, and is receiving increasing attention from researchers in industry and academia.

At present, the mainstream 3D human pose estimation algorithms can be divided into traditional and deep learning methods. In the early days, traditional human pose estimation methods generally included a detector of human body parts that was based on a graph structure and a deformable part model, and described the spatial relationships between the body parts through a graph model. The extracted features were mainly set manually, such as HOG, SIFT and other features [2]; these approaches could not make full use of the image features, and were susceptible to image appearance. Moreover, the part model had a single structure, which imposed limitations on the possible application scenarios.

With the advent of the era of big data, deep learning approaches have been rapidly developed for the field of computer vision, and research on human pose estimation has shifted from traditional methods to deep learning methods. Deep-learning-based pose estimation methods mainly use convolutional neural networks (CNNs) to extract human pose features from images, thus avoiding the need for the manual design of image features. In 2014, Li et al. [5] first showed that deep neural networks could achieve reasonable accuracy in 3D human pose estimation from a single image. Their framework included joint regression networks and body part detectors. Although many approaches had previously been proposed to try to predict a 3D pose directly from an image, Chen et al. [6] explored a simple approach to 3D human pose estimation by performing 2D pose estimation followed by 3D exemplar matching.

Although existing methods have achieved remarkable results, there are still significant challenges in regard to 3D human pose estimation at the application level. Since the most commonly used 3D datasets were captured by motion capture systems (MOCAPs) in controlled laboratory environments, the cost of 3D ground truth labels is high, and the background of the image is single. Network models trained on these datasets cannot easily be extended to outdoor and complex environments with significant occlusion of body parts. However, data without 3D labels are easily available, so the question arises as to whether we can train a network model using data without 3D ground truth labels. To solve this problem, we construct a network model for 3D human pose estimation in this paper, and use a self-supervised training method to estimate 3D human pose accurately from images without 3D labels.

We present a 3D human pose estimation network model called Pose ResNet, which integrates the WASP and CBAM modules into the base network of ResNet50. We use epipolar geometry to construct 3D labels to realize self-supervision, hence solving the problem of obtaining 3D ground truth labels for datasets for 3D pose estimation technology. Our model is pre-trained on a 2D dataset and then undergoes further self-supervised training on a 3D dataset without 3D labels after migrating the trained model parameters, and synthetic occlusion is used to enhance the training images. Without the need for 3D ground truth labels, the accurate estimation of 3D human pose is realized from 2D images. 

The contributions of this study are as follows:(1)A network model called Pose ResNet is designed for 3D human pose estimation. This model is based on ResNet50 and WASP, and a CBAM attention mechanism is introduced to increase the receptive field and select the important pixels in a fine-grained way.(2)This model uses the strategies of transfer learning and data enhancement for training, which not only improve the accuracy of joint detection but also enhance the generalization performance of the model. Transfer learning involves a transfer from 2D to 3D data, while data enhancement refers to the synthetic occlusion of images in the 3D dataset.(3)A set of 3D labels constructed through an epipolar geometric transformation between multiple views are used to train this model. Without the need for 3D ground truth labels, this model realizes true self-supervised training.

The rest of this paper is organized as follows: In Section 2, related work on 3D human pose estimation is described. In Section 3, the proposed model and method are introduced. In Section 4, the datasets and training process of the model are introduced. In Section 5, we explain the evaluation index used, and present experimental results, an analysis and a discussion. In Section 6, we summarize our results and discuss prospects for future work.

## 2. Related Work

In this paper, a single-stage method is used to estimate 3D pose directly from an image, and self-supervised method is used in training. This section will give an overview of the deep learning method of human pose estimation from two different perspectives: the number of model stages and the supervision mode.

### 2.1. Single- and Two-Stage Models

In recent years, CNNs have made significant achievements in the field of image processing, and deep learning technology has been used to meet the need for pose estimation in practical applications more effectively. Three-dimensional human pose estimation methods based on deep learning are mainly divided into two categories based on the number of stages: single-stage methods, which train the neural network to regress joint locations directly, and two-stage methods, which decompose the 3D pose estimation task into two independent stages, involving the estimation of 2D poses and then lifts them into 3D space.

Many papers have adopted single-stage methods, such as [7,8,9,10,11,12,13], in which CNNs were used to estimate the coordinates of 3D joints directly from images. Li et al. [5] combined a regression network with a body part detection network, and for the first time proved that a deep neural network could directly obtain a 3D human pose from a single image. The estimation accuracy was significantly improved compared with traditional methods. Tekin et al. [8] trained a self-encoder–decoder by combining traditional CNNs for supervised learning with auto-encoders for structural learning and achieved good results. Apart from that, Pavlakos et al. [14] described the problem of 3D human pose estimation as 3D keypoint localization in a discretized space around the subject. The advantage of a single-stage method is that the whole network model can achieve end-to-end training, although the requirements for the network structure are higher.

Compared to single-stage methods, which directly regress 3D coordinates from images, two-stage methods can also achieve good results. Zhou et al. [15] proposed a transfer learning method that used a mixture of 2D and 3D labels in a deep neural network based on a two-stage cascaded structure. Their network was augmented with a state-of-the-art 2D pose estimation sub-network with a 3D deep regression sub-network. In addition, an advanced 2D pose estimation module can be directly used to obtain 2D poses. By inputting 2D poses into a 3D pose estimation module, 3D poses can be obtained [16,17,18]. Martinez et al. [16] built a system that could predict 3D joint positions given 2D joint positions, and suggested directions for future work that could further advance the state of the art in 3D human pose estimation. Compared with single-stage methods, this approach benefits from the maturity of the current 2D pose estimation technology [19,20,21], and reduces the learning pressure on the model.

### 2.2. Weakly Supervised and Self-Supervised Methods

The successful application of deep learning in computer vision relies heavily on large amounts of fully labeled data, although labeled data with high confidence are sometimes difficult to obtain. Due to the high cost of acquiring 3D annotations for 3D pose estimation datasets, weakly supervised and self-supervised methods have become research hotspots in recent years.

Many researchers [22,23,24,25,26] have explored methods of human pose estimation based on weak and self-supervision. Compared with fully supervised methods [8,9,10,11,12,27] in which regression networks are used to estimate 3D joint coordinates directly from images, self-supervision and weakly supervised methods can solve the difficulty of obtaining 3D ground truth labels to a certain extent. Pavlakos et al. used [26] a pictorial structure model to obtain global pose configurations from key point heatmaps of multi-view images. Nevertheless, their method needed full camera calibration and a keypoint detector to produce the 2D heatmaps. Rhodin et al. [24] used multi-view consistency constraints to supervise a network. They needed small amounts of 3D ground truth data to avoid degenerate solutions in which the pose collapses to a single location. Thus, a lack of in-the-wild 3D ground truth data represent a limiting factor for this method. Drover et al. [23] used an adversarial framework to impose a priori on 3D structures, learning only from their random 2D projections. Their method did not require correspondence between the 2D and 3D points to construct explicit 3D priors, but still requires the use of 3D labels. Zhou et al. [15] mixed 2D data with 3D indoor data to form a batch. For the 3D data, a Euclidean loss function was used, while for the 2D data, a weak supervision loss was proposed based on the 2D annotations and prior knowledge of the human skeleton. Their approach, while enhancing the applicability of outdoor images, still requires 2D and 3D labels. Wandt et al. [28] trained a generative adversarial network that consisted of three parts: a pose and camera estimation network, a critic network and a reprojection network. They used an additional camera estimation network and a reprojection layer that projected the estimated 3D pose back to 2D. Kocabas et al. [29] showed that even without any 3D ground truth data or a knowledge of camera external paramenters multi-view images could be leveraged to achieve self-supervision. Hua et al. [30] proposed a coarse-to-fine approach for cross-view 3D human pose estimation. Specifically, they first use triangulation to lift the 2D detections to coarse 3D poses, and then use a refined model to obtain accurate results. Although their method is more accurate, they need to input multi-view images in both the training stage and the estimation stage, while our method only needs to input a monocular image in the estimation stage to estimate the 3D pose. Gong et al. [31] proposed a new self-supervised method: PoseTriplet. Their model consists of three parts: pose estimator, imitator and illusionist. The pose estimator transforms an input 2D pose sequence to a low-fidelity 3D output, which is then enhanced by the imitator that enforces physical constraints. The refined 3D poses are subsequently fed to the hallucinator to produce even more diverse data. Although their method does not rely on any given 3D data, their network uses 2D pose sequence data as input and cannot achieve direct estimation of 3D pose from 2D images.

Existing 3D human pose estimation methods usually require 3D labels or camera internal and external parameters and rely heavily on consistency or adversarial loss to generate supervised signals, which are weak and can affect the performance of the model. Unlike the above existing methods, the supervised signal of our method is a strongly supervised signal, which is directly supervised with pseudo-labels calculated by an epipolar geometric formula, without using 3D labels or external parameters of the camera, and the trained network model is capable of 3D human pose estimation from monocular images.

## 3. Models and Methods

As a 3D human pose estimation model based on self-supervision, Pose ResNet is a dual-branch structure consisting of upper and lower branches. As shown in Figure 1, the upper branch is composed of a 2D Pose CNN and an epipolar geometric module, and the lower branch is composed of a 3D Pose CNN. When the 2D Pose CNN of the upper branch has output the 2D human body pose, the 3D label is constructed by using the epipolar geometric module to supervise the training of the 3D Pose CNN of the lower branch, thus realizing self-supervision.

### 3.1. Pose ResNet

As shown in Figure 1, the 2D Pose CNN in the upper branch is composed of a feature extraction module and key joint detection module, while the 3D Pose CNN in the lower branch is also composed of a feature extraction module and key joint detection module. The feature extraction modules of both branches are the same, and ResNet50 is used as the basic network to extract features. The ResNet network is simple and practical, and has been widely used in various feature extraction. When the number of layers of deep learning network is deeper, theoretically the network performance will be stronger, but after the CNN network reaches a certain depth, and then deeper, the network performance will not improve, but it will lead to a slower convergence of the network, and the accuracy rate decreases with it. Additionally, the skip connection of the ResNet network can solve this problem well and alleviate the gradient disappearance to some extent. The CBAM is introduced before Layer 1 and after Layer 4 of ResNet50 for the fine-grained selection of important pixels. The WASP module is added after the ResNet-CBAM, which is applied to the extracted features to increase the receptive field and capture the multi-scale context information of the image. Finally, a deconvolution network is connected after the WASP module, and the features are applied to the deconvolution network to obtain the volumetric heatmap H.

The key joint detection module of the 2D Pose CNN obtains the 2D pose U by applying a soft argmax function to the x- and y-dimensions of the volumetric heatmap H. The key joint detection module of the 3D Pose CNN directly obtains the predicted 3D pose V by applying a soft argmax function to the x-, y- and z-dimensions of the volumetric heatmap H.

The self-supervised learning of this model involves mining and constructing the supervision information from the data without manual annotations, and training the network based on the supervision information. Self-supervision in this case is achieved by the epipolar geometry module of the upper branch. As shown in Figure 1, the 2D Pose CNN in the upper branch outputs the multi-view 2D pose, and the epipolar geometry module then constructs the 3D pose and caches it as a 3D label $\hat{V}$. The constructed 3D labels are used for self-supervised training of the 3D Pose CNN in the lower branch. To some extent, this solves the difficulty of obtaining 3D ground truth labels for 3D human pose datasets.

### 3.2. CBAM

The attention mechanism module operates similarly to the selective visual attention mechanism of humans. The aim is to select information that is more important to the current target task from all of the information that is available, thus effectively suppressing useless information and greatly improving the efficiency and accuracy of visual information processing. In the estimation of a human pose, due to the presence of complex image backgrounds, the spatial location information of the human pose features in the image plays an important role in accurately identifying the joints. Attention mechanisms can be used to suppress redundant background information and select the important pixels in a fine-grained way, thus making the network pay more attention to the joint information of human body. In this study, a CBAM [32] that combines channel and spatial attention is applied to the output feature map of the convolution network to improve the estimation performance of the model.

In this paper, the CBAM attention mechanism, an attention mechanism module that combines spatial and channel information, is introduced before Layer 1 and after Layer 4 of the basic ResNet50 network [32]. The structure of ResNet50 after the introduction of the CBAM module is shown in Figure 2.

As shown in Figure 3, the CBAM module contains two submodules cascaded in sequence: a channel attention (CA) submodule and a spatial attention (SA) submodule. As shown in Figure 3a, the input feature map is first passed through the CA module to get the initial weighted results, and then through the SA module to get the final weighted results.

The CA submodule weights the channel dimensions of the input feature map, its structure is shown in Figure 3b. We assume that the size of input feature map F is h×w×c where w is the width of the feature map, h is the height and c is the number of channels. The CA module can be expressed by the mathematical equation
(1)$$M_{C}\left(F\right)=\sigma \left\{MLP\left[AvgPool\left(F\right)\right]+MLP\left[MaxPool\left(F\right)\right]\right\}$$
where AvgPool and MaxPool represent average pooling and max pooling, respectively, which give two sets of feature vectors of size 1×1×c. MLP represents a learnable multilayer perceptron with one hidden layer, which can output a set of learned feature vectors. σ is the sigmoid activation function used to map each point of the feature vector to between zero and one, and $M_{S}\left(F\right)$ are the weights of the different channels.

The SA submodule weights the spatial positions of the input feature map, its structure is shown in Figure 3c, and can be expressed by the mathematical equation
(2)$$M_{S}\left(F\right)=\sigma \left\{f^{7\times 7}\left[AvgPool\left(M_{C}\right);MaxPool\left(M_{C}\right)\right]\right\}$$
where $f^{7\times 7}$ represents a convolution kernel with size 7 × 7, which is the spatial feature extractor of the target. The feature map is max pooled and average pooled along the channel dimension. The SA submodule uses the sigmoid activation function after the 7 × 7 convolution to map the value of each pixel in the feature map to a probability value between zero and one. 

### 3.3. WASP Module

In order to capture the multi-scale context information of the image, expand the receptive field and make the detection of joint position more accurate, the WASP module [33] is added to the model. The receptive field refers to the area size of the input layer corresponding to an element deciding the output result of a certain layer. When the input image is passed through the different convolutional layers of the network, the structure of the image that can be seen is different. After passing through a shallow convolution layer, each pixel represents only one feature extraction of the local information of the original image, with rich detail but little context information, meaning that the receptive field is small. In contrast, after passing through a deep convolutional layer, the range of the original image can be seen is larger, but some details are missed, meaning that the receptive field is large.

Hence, occlusion is a difficult problem when estimating the 3D pose of the human body. Expanding the receptive field enables the network to learn the relationships between the joints of the body using context information, which is effective for the inference of occluded joints. However, the differentiation between local information of the joints in the background is weak, which can easily cause local confusion. Expanding the receptive field makes the range of the visible image larger and enhances the differentiation of the local information on the joints.

Before applying the WASP module, we first introduce atrous convolution, which is a way of increasing the receptive field. Atrous convolution means injecting atrous into the standard convolution to increase the receptive field. As shown in Figure 4, Figure 4a is the schematic diagram of standard convolution and Figure 4b is the schematic diagram of atrous convolution. Compared with the standard convolution, atrous convolution has an extra parameter called dilation rate, which refers to the interval number of kernels. The specific meaning is to fill 0 in the middle of the convolution kernel, and the number of filled 0s is the dilation rate −1. In this case, we set the kernels to 3, and the dilation rate to 2 (a dilation rate of 1 is the standard convolution).

The WASP module samples the input in parallel with empty convolutions of different sampling rates and combines the results together. The number of channels is then reduced to the expected value through 1 × 1 convolution, as shown in Figure 5, which is equivalent to capturing the context information of the image in multiple proportions. The addition of the WASP module means the network can make better use of global information, rather than paying too much attention to a small proportion of the features. In addition to enriching the level of detail, it also expands the receptive field to ensure that important information is not ignored when making decisions.

## 4. Model Training

### 4.1. Dataset

At present, the lack of large outdoor datasets and datasets with special types of poses (such as falling, rolling, etc.) constitutes the main bottleneck to the development of 3D pose estimation models. This phenomenon mainly arises because 3D datasets are built with MOCAP systems, which are suitable for indoor environments but require complex devices with multiple sensors and tights that are impractical for outdoor use. In this paper, we propose a solution to this problem by using two large public datasets, namely an outdoor dataset called MPII [34] and an indoor 3D dataset called Human3.6M [35]. The model is pre-trained on the former and then undergoes self-supervised training on the latter.

MPII [34], a dataset used for evaluating 2D human pose estimation, contains about 25,000 labeled images with 16 annotated joints. These images are extracted from YouTube videos. Overall, this dataset covers 410 kinds of human behaviors, most of which are captured in outdoor environments with complex backgrounds, as shown in Figure 6, which shows some images of this dataset.

The Human3.6M dataset [35] is the largest and most widely used dataset for 3D human pose estimation, and contains 3.6 million RGB images from four different perspectives captured by MOCAP systems in indoor environments. The dataset includes 11 subjects in 15 action scenarios, such as talking, eating, exercising, greeting, etc., as shown in Figure 7a. The dataset includes annotations for 17 joints of the human body, and the topology of the joints is shown in Figure 7b. Although 11 subjects participated in the dataset, 3D annotations are available for only seven (S1, S3, S5, S7, S8, S9, S11). In general, subjects S1, S3, S5, S7 and S8 are used as the training set, while S9 and S11 are used as the testing set. To reduce data redundancy, key frames are extracted from the videos. In the training set, one frame is extracted from every 5 frames, while in the testing set, one frame is extracted from every 64 frames. After preprocessing to remove redundancies, there are 51,765 samples in the training set and 8606 in the testing set.

The MPII dataset was used for pre-training, and the Human3.6M dataset was then used for self-supervised training. In the MPII dataset, the skeleton topology has 16 joints connected according to certain relationships, while in the Human3.6M dataset, the skeleton topology has 17 joints. The skeleton topology finally output by the network model has 16 joints, and joint 7 (spine) in Figure 7b is abandoned.

### 4.2. Data Augmentation

At the training stage, synthetic occlusion [36] was used to enhance the images in the Human3.6M dataset. In the dataset without occlusion, all targets are clearly visible, and the trained network may achieve relatively high accuracy. However, the generalization ability of the model is poor, the accuracy may be low for occluded objects and some targets may not even be recognized. Synthetic occlusion was therefore used to enhance the training images so that the network could make better use the of global information, rather than paying too much attention to a small part of the features, in order to improve the robustness of the model.

Synthetic occlusion was applied using the Pacal VOC dataset. Segmented objects extracted from the Pacal VOC dataset were filtered to remove people and objects marked as difficult or truncated, and the remaining 2638 objects were pasted into random positions in the Human3.6M dataset with a certain probability (P_occ_) to synthesize training images with random occlusion [36]. During the experiment, the probability of occlusion (P_occ_) was set to 0.5, and the degree of occlusion was between 0% and 70%. Examples of synthesized occluded images are shown in Figure 8, where the original image is shown on the left and the occluded images are shown on the right. The TV, bus, dog, bird, etc., in Figure 8 are objects extracted from the Pacal VOC dataset.

### 4.3. Pre-Training of 2D Pose CNN

At the training stage of the 3D human pose estimation model, the 2D Pose CNN was first pre-trained on the MPII dataset, so that the network could achieve accurate estimation of the 2D human pose. The trained parameters were then transferred to the 3D Pose CNN, and the Human3.6M dataset was used for self-supervised training. The data used for training were synthetically occluded. The details of the training process are shown in Figure 9.

Pre-training is one application of transfer learning. After pre-training, the model can achieve an accurate estimation of the 2D pose, which makes the 3D labels that are constructed based on the 2D pose more accurate, thus improving the accuracy rate of joint detection. Furthermore, the environments of the images in the 2D datasets are mostly outdoors with complex backgrounds, while the 3D datasets are mostly acquired in indoor closed environments by MOCAP systems with monotonous backgrounds. Hence, better results can be obtained by pre-training on the MPII dataset and then carrying out self-supervised training on the Human3.6M dataset.

### 4.4. Self-Supervised Training of 3D Pose CNN

In self-supervised training, the supervised information is not manually annotated, but it is automatically constructed by algorithms from unsupervised data. The use of self-supervised learning techniques can reduce the dependence of network models on data, which solves the problem of the difficulty of acquiring 3D annotations. The details of the implementation of self-supervised learning in this paper are shown in Figure 1. The 3D labels are calculated from the 2D pose of the multi-views output from the upper branch on the model by the epipolar geometric formulation.

For self-supervised training, we assume that the number of views is two. The different views images *I_i_* and $I_{i+1}$ with resolution 1000 × 1000 from the Human 3.6M dataset are used as input to both the upper and lower branches of the model. The 2D Pose CNN in the upper branch directly outputs the multi-view 2D poses $U_{i}$ and $U_{i+1}$. The epipolar geometric transformation of $U_{i}$ and $U_{i+1}$ is then carried out to obtain the 3D pose $\hat{V}$ in the global coordinate frame, and this is cached as the 3D label. The 3D Pose CNN in the lower branch outputs the 3D human pos *V* predicted from input images. It then calculates the loss between V and $\hat{V}$. To calculate the loss, we project V onto the corresponding camera space, and then minimize ${smooth}_{L1}\left(V-\hat{V}\right)$ to train the lower branch, where
(3)$${smooth}_{L1}\left(x\right)=\{\begin{array}{ccc} 0.5x^{2} & \; & if\left|x\right|<1\\ \left|x\right|-0.5 & \; & otherwise \end{array}$$

We can substitute $x=V-\hat{V}$ into Equation (3) to calculate the loss, where V is the 3D pose predicted by the lower branch in camera space and $\hat{V}$ is the 3D label obtained from epipolar geometry in the upper branch.

Epipolar geometry describes the internal photographic relationship between the two views, which is independent of the external scene environment, and is only related to the internal parameters of the camera and the relative pose between the two views. The upper branch of the model uses epipolar geometry to obtain the 3D labels $\hat{V}$, using the following steps:(a)Let the 2D coordinate of the j joint in the i image be $U_{i,j}=\left[x_{i,j},y_{i,j}\right]$ and its 3D coordinate be $V_{j}=\left[X_{j},Y_{j},Z_{j}\right]$. We can then describe the relation between them by assuming a pinhole image projection model. The following formulas are obtained from the pinhole image projection model:(4)$$U_{i,j}FU_{i+1,j}=0$$
(5)$$E=K^{T}FK$$
(6)$$\left[\begin{array}{c} x_{i,j}\\ y_{i,j}\\ w_{i,j} \end{array}\right]=K\left[R\left|RT\right.\right]\left[\begin{array}{c} X_{j}\\ Y_{j}\\ Z_{j}\\ 1 \end{array}\right],K=\left[\begin{array}{ccc} f_{x} & 0 & c_{x}\\ 0 & f_{y} & c_{y}\\ 0 & 0 & 1 \end{array}\right],T=\left[\begin{array}{c} T_{x}\\ T_{y}\\ T_{z} \end{array}\right]$$
where $\omega_{i,j}$ is the depth of the j joint in the i camera’s image with respect to the reference frame of the camera, K encodes the intrinsic parameters of the camera ($f_{x}$ and $f_{y}$ are the focal length,$c_{x}$ and $c_{y}$ are the principal points), and *R* and *T* are the extrinsic parameters of rotation and translation of the camera.(b)Without using the external parameters of the camera, we can assume that the first camera is located at the center of the coordinate system, meaning that *R* for this camera is constant. The corresponding joints in $U_{i}$ and $U_{i+1}$ satisfy Equation (4). By substituting $U_{i}$ and $U_{i+1}$ into Equation (4), the fundamental matrix *F* can be obtained.(c)By substituting *K* and *F* into Equation (5), we obtain the essential matrix *E*. By using *SVD* to decompose *E*, we get four possible solutions to *R*.(d)By substituting the 2D coordinates into the epipolar geometry triangulation in Equation (6), we obtain the 3D coordinates of the corresponding joints by polynomial triangulation, and cache them as 3D labels, $\hat{V}$.

## 5. Experiments

### 5.1. Evaluation Indicators

The percentage of correct keypoints (PCK) [34] is the most widely used evaluation index for 2D pose estimation. PCK is defined as the proportion of correctly estimated joints, when the normalized distance between the predicted coordinates of the joint and the ground truth coordinates is less than a pre-set threshold, the estimation of the joint is considered correct. Otherwise, the estimation is considered incorrect. The formula for calculating PCK is shown in Equation (7), where *i* represents the number of joints, $d_{i}$ represents the Euclidean distance between the predicted value and the real value of the i joint, *d* is the scale factor of the human body and *T* is the threshold value. The normalized distance is the Euclidean distance between the predicted value of the joint and its corresponding ground truth normalized by the human scale factor. The normalization of the MPII dataset uses the current human head diameter as the scale factor, and the PCK threshold set to 0.5 in the paper means that the Euclidean distance between the detected point and its corresponding ground truth ≤ 0.5*head diameter is considered when the estimation of the joint is correct. For different datasets, the normalized scale factors may be different.
(7)$$PCK=\frac{\sum_{i}\delta \left(\frac{d_{i}}{d}\leq T\right)}{\sum_{i}1}$$

MPJPE [35] is the most widely used evaluation index for 3D pose estimation. It is obtained by calculating the average value of the Euclidean distance between the estimated coordinates and the real coordinates of all the joints. The formula is shown in Equation (8), where $m_{f,s}^{\left(f\right)}(i)$ is the predicted value of the i joint of bone *s* in the f frame, $m_{gt,s}^{\left(f\right)}(i)$ is the real value of the i joint of the bone *s* in the f frame, $N_{s}$ is the number of joint points of bone *s* and “||.||_2_” means the L2 norm, that is, the Euclidean distance. Before evaluation, the estimated and real coordinates of the root joint were aligned via rigid body transformations such as translation and rotation.
(8)$$E_{MPJPE}\left(f,s\right)=\frac{1}{N_{S}}{\sum_{i=1}^{N_{S}}\left\|m_{f,s}^{\left(f\right)}(i)-m_{gt,s}^{\left(f\right)}(i)\right\|}_{2}$$

### 5.2. Experimental Environment and Parameter Selection

The experimental environment was a Linux system, and the experiment was carried out on an NVIDIA 2070 (8G) graphics card. The network models were built in Pytorch, and the ADAM optimizer was used to optimize the model. The initial learning rate was set to 0.001, the batch size was 16 and the number of iterations (epochs) was 130.

### 5.3. Analysis of Experimental Results

In this paper, we present a self-supervised 3D human pose estimation model. An example of the estimation results on the Human3.6M dataset is shown in Figure 10. From left to right of Figure 10, there are the original input images, the skeleton graphs of the 2D pose and the skeleton graphs of the 3D pose. The epoch-loss plot of the model training process is shown in Figure 11.

To validate the accuracy of the 3D human pose estimation model based on self-supervision in this paper, comparative experiments were carried out from the following four aspects.

#### 5.3.1. Comparison with Fully Supervised Methods

In order to demonstrate the advantages of the proposed method, it was compared with other existing fully supervised methods. A fully supervised method is one that uses the ground truth labels of the dataset during training. Table 1 shows a comparison of the experimental results from our method with other fully supervised methods on the Human3.6M dataset, and the best of the evaluation metrics are highlighted for better reading. Compared with the others, our method has a simpler structure, fewer parameters, a smaller MPJPE and a higher accuracy of 3D pose estimation. Although Table 1 shows that the MPJPE of Zheng et al.’s methods [37,38] are smaller than the MPJPE of our fully supervised method, their methods require input video information. Furthermore, although our self-supervised method is not excellent compared with the other fully supervised methods, it does not require any 3D ground truth labels, and the 3D human pose estimation is realized based on 2D image data.

#### 5.3.2. Comparison with Weakly Supervised and Self-Supervised Methods

Due to the high cost of 3D annotation, it is very difficult to obtain strong supervised information in many cases, and weakly supervised and self-supervised methods [28,29,39,40,41,42] have therefore become very popular in recent years. There is no clear boundary between self-supervision and weak supervision, both of which aim to construct some supervised information to obtain pseudo-labels from unlabeled data. Table 2 compares the experimental results of our method with those of other weakly and self-supervised methods on Human3.6M data. Although Table 2 shows that the method of Lqbal et al. has a minimum error of 69.1 mm, their model uses multiple views for prediction, meaning that multiple images of different views need to be input for the prediction of human body pose, while our method only needs one single image to be input to obtain the 3D pose after training is completed. For practical applications, 3D pose estimation based on monocular images is more widely used and is more convenient than 3D pose estimation based on multiple views. Overall, we see that the method proposed in this paper can accurately predict 3D pose estimation from monocular RGB images without the need for 3D ground truth labels.
sensors-23-03057-t001_Table 1Table 1Comparison of results between the proposed method and other fully supervised methods.ModelMPJPE (↓)P_MPJPE (↓)N_MPJPE (↓)Nie et al. [27] (2017)97.579.5
Rhodin et al. [24] (2018)66.851.663.3Martinez et al. [16] (2017)62.947.7―Pavlakos et al. [14] (2017)71.9――Sun et al. [11] (2018)49.640.6―Kocabas et al. [43] (2020)65.641.4―Kevin Li et al. [44] (2021)54.036.7―Zheng et al. [37] (2021)44.334.6―Zhang et al. [38] (2022)40.932.6―Ours (fully supervised)51.841.949.4Ours (self-supervised)74.663.873.2


#### 5.3.3. Comparison of Different Depth Networks

In order to verify the influence of different network depths, we compared the experimental results from residual networks with different depths at the pre-training stage. The higher the accuracy of 2D pose estimation at the pre-training stage, the better the results after migration and the closer the pseudo-label calculated by the epipolar geometry to the ground truth label. Hence, the accuracy of pre-training has a significant influence on the final experimental results. As shown in Table 3, the PCK values for ResNet18, ResNet34, ResNet50 and ResNet101 are 84.7%, 86.3%, 88.3% and 88.9%, respectively. It can be observed that the precision of the ResNet101 model is the highest, followed by ResNet50. However, since the network depth and the number of parameters in ResNet101 are far greater than those of ResNet50, ResNet50 was finally used as the basic network for the feature extraction module in this paper.

#### 5.3.4. Ablation Experiment

In order to verify the effects of transfer learning, the use of occlusion to enhance the training data, and the introduction of the CBAM and WASP modules to the network, a set of ablation experiments was designed. The results are shown in Table 4. We note that CBAM is an attention mechanism that combines channel attention with spatial attention, WASP is a spatial pooling module based on a waterfall model, and it can be integrated with feedforward CNNs.

It can be seen from Table 4 that the use of transfer learning, i.e., pre-training on the 2D dataset and then transferring to the 3D dataset for further training, had a great influence on the accuracy. Compared with the model without transfer learning, the MPJPE was reduced by 19 mm. In addition to transfer learning, the use of occlusion also had a strong influence on the experimental results. Compared with the model using transfer learning alone, the MPJPE for the model with synthetic occlusion after transfer learning was reduced by 7.6 mm. The use of CBAM and WASP modules further reduced the MPJPE of the model by 1.5 mm.

### 5.4. Discussion

Figure 12 shows an example of a heatmap of each joint of the model in an image from the Human3.6M dataset, from which it can be seen that some of the joints of the human body were estimated accurately, while others had obvious errors. Figure 13 presents a histogram of the MPJPE values for the human joints, which more intuitively show the differences in accuracy between the different joints. As can be seen from Figure 13, the MPJPE values for the ankle, neck, head and wrist were relatively high, while the value for the hip joint was the lowest, and the values for the corresponding joints on the left and right parts of the body were basically the same. From the perspective of human joint biomechanics, in many actions, the ankle and wrist joints are far from the central joint, which leads to high complexity for the action and hence some degree of overlap and occlusion. For example, for the action of “sitting down”, the overlap and occlusion of the ankle joint were the most serious. However, the hip joint was close to the central joint, so the complexity of the action was low and the accuracy was the highest. The MPJPE for the hip joint was zero, as this was taken as the central joint when calculating the MPJPE.

## 6. Conclusions and Prospects

Given that obtaining 3D data for human pose estimation is a challenging problem, this paper proposes a method to solve this problem by using epipolar geometry to construct 3D labels for self-supervised learning, achieving excellent results. The network uses ResNet50 as the basic network in the feature extraction module. In order to learn more fine-grained features, a CBAM module combining channel attention and spatial attention is also introduced into the ResNet50. For the extracted features, the WASP module is used to capture multi-scale context information from the image to increase the receptive field. The model also uses two learning strategies, transfer learning and synthetic occlusion. The final MPJPE on the Human3.6M dataset was reduced to 74.6 mm.

In future research, the model will be further optimized for the small target problem in order to improve the applicability to natural scenes, in which the area represented by the person in the image is too small or the subject is not in the middle of the image.

## Figures and Tables

**Figure 1 sensors-23-03057-f001:**
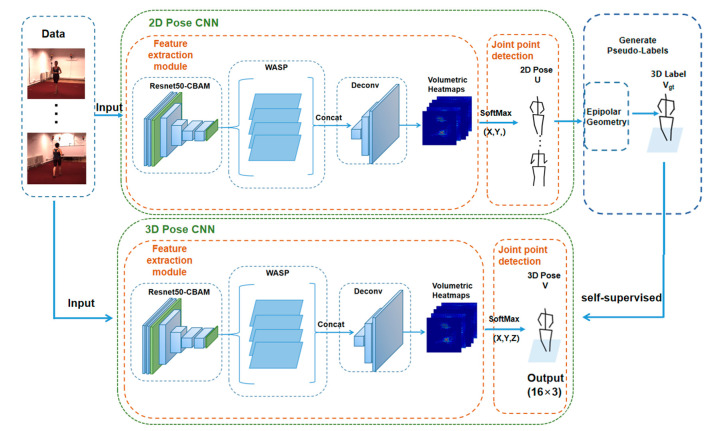
Structure of Pose ResNet.

**Figure 2 sensors-23-03057-f002:**
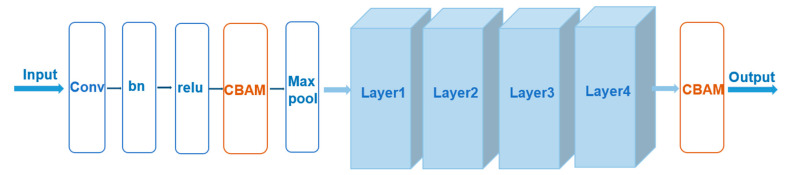
Structure diagram of ResNet50-CBAM.

**Figure 3 sensors-23-03057-f003:**
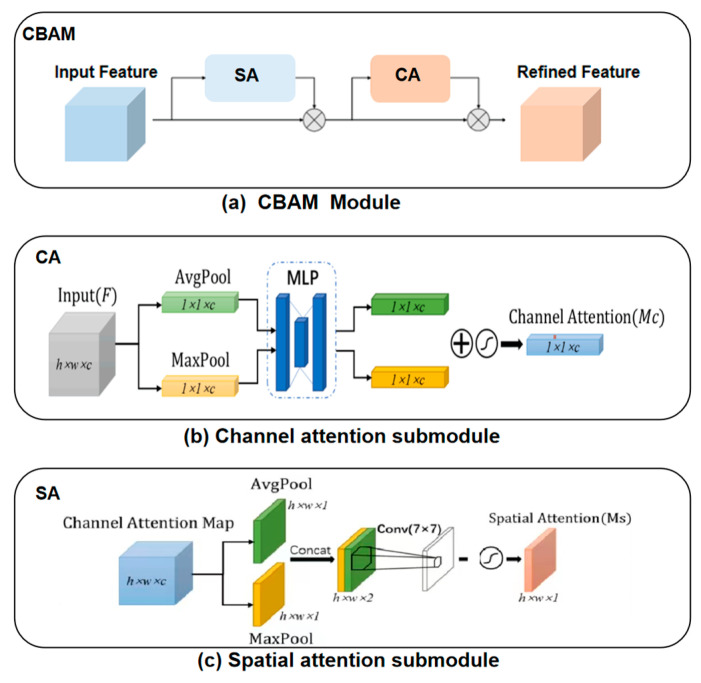
Structure of the CBAM module.

**Figure 4 sensors-23-03057-f004:**
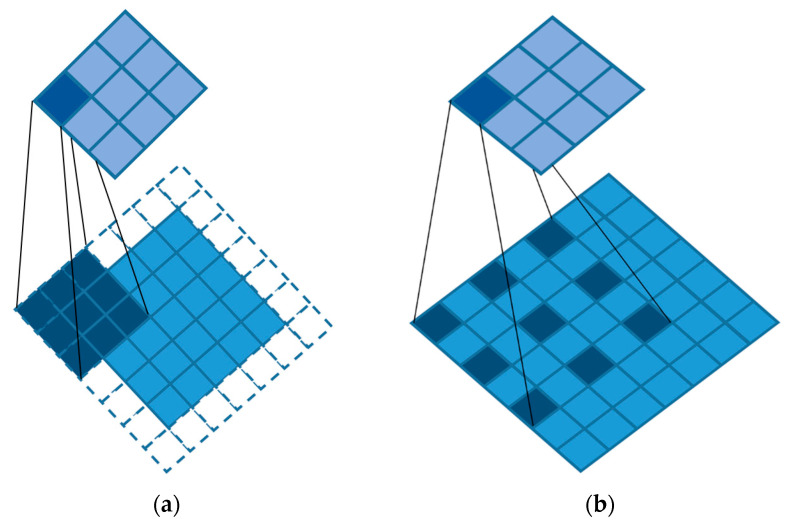
(**a**) Standard convolution with a 3 × 3 kernel. (**b**) Atrous convolution with a 3 × 3 kernel.

**Figure 5 sensors-23-03057-f005:**
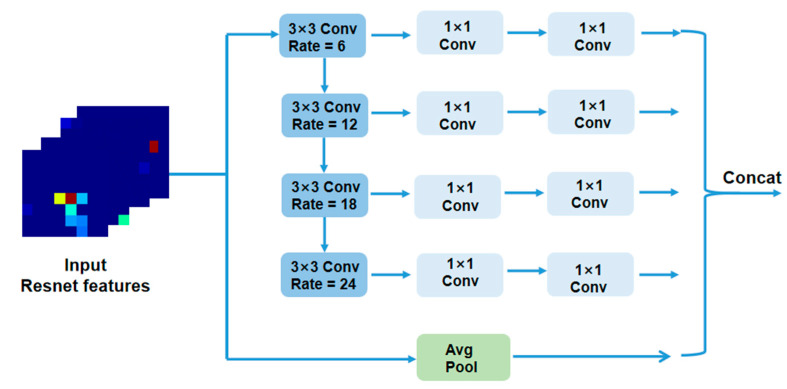
Structure of the WASP module.

**Figure 6 sensors-23-03057-f006:**
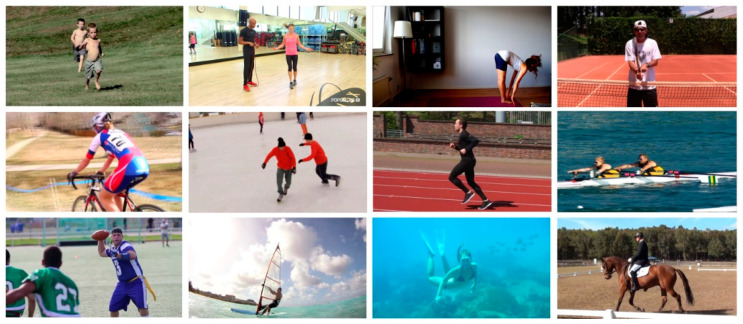
Images from the MPII dataset.

**Figure 7 sensors-23-03057-f007:**
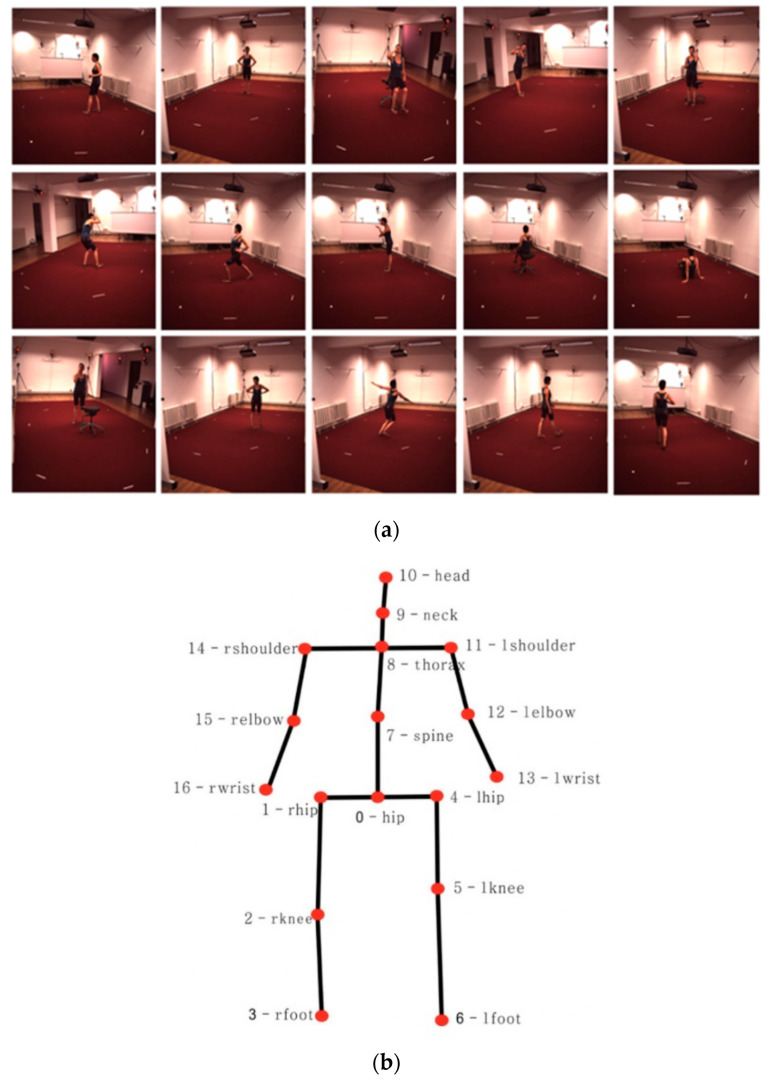
(**a**) Images from the Human 3.6M dataset; (**b**) skeleton topology in the Human3.6M dataset.

**Figure 8 sensors-23-03057-f008:**
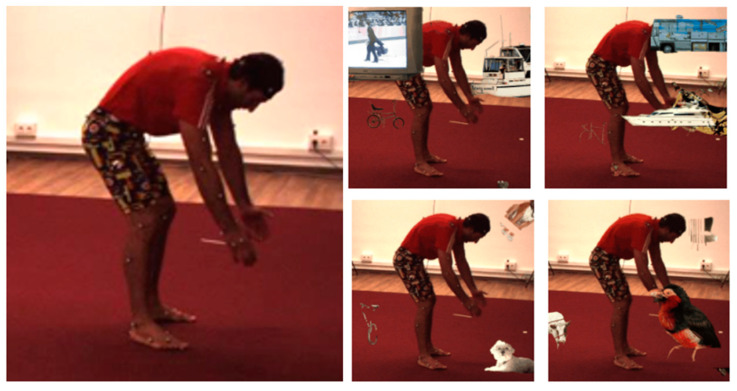
Occluded images from the Human3.6M dataset.

**Figure 9 sensors-23-03057-f009:**
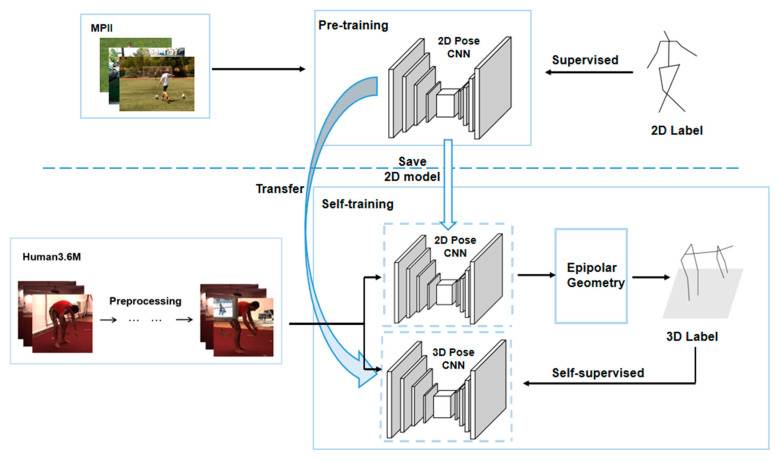
Training process of Pose ResNet.

**Figure 10 sensors-23-03057-f010:**
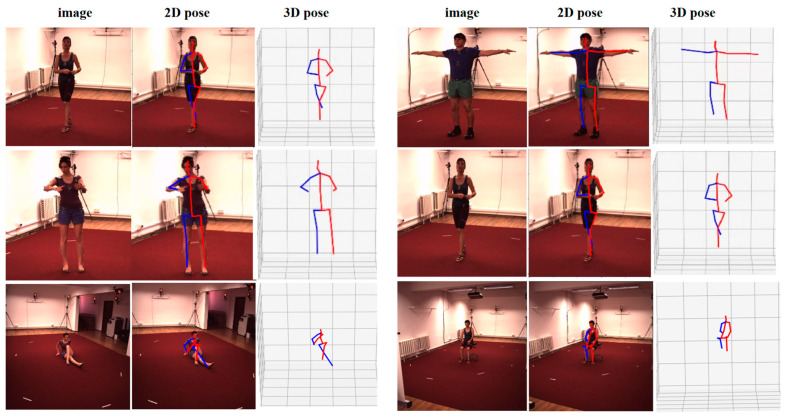
Examples of estimation results on the Human3.6M dataset (from **left** to **right**, the columns show the original image, 2D pose, and 3D pose).

**Figure 11 sensors-23-03057-f011:**
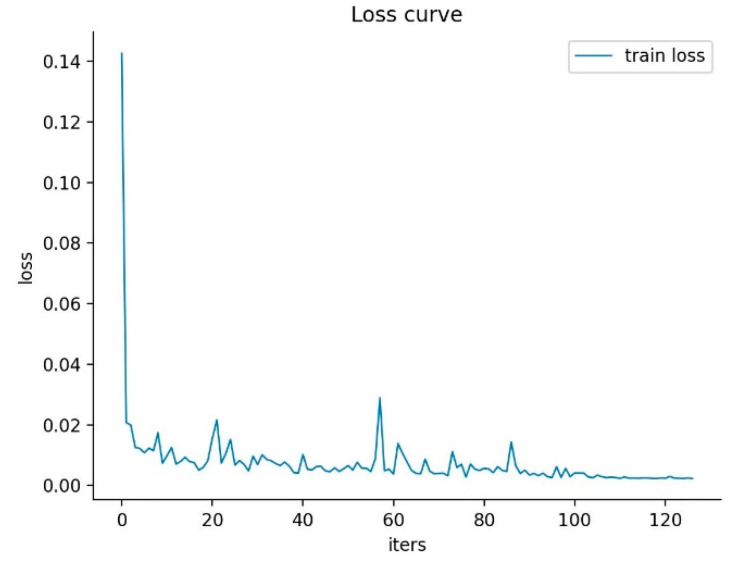
Epoch-loss plot of the model training process.

**Figure 12 sensors-23-03057-f012:**
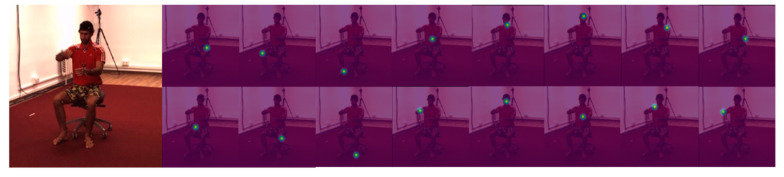
Heatmap visualization of each joint of the model in an image from the Human3.6M dataset.

**Figure 13 sensors-23-03057-f013:**
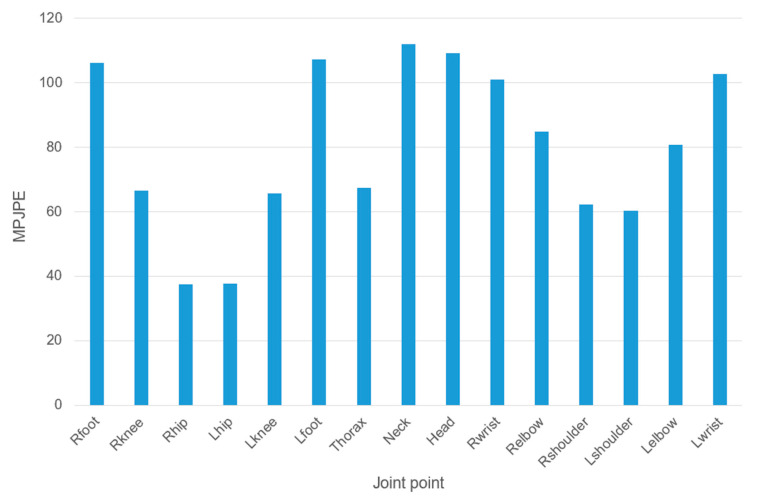
MPJPE for each joint for our method on the Human3.6M dataset.

**Table 2 sensors-23-03057-t002:** Comparison of results between the proposed method and other weakly supervised and self-supervised methods.

Model	MPJPE (↓)	P_MPJPE (↓)	N_MPJPE (↓)
Pavlakos et al. [14] (2017)	118.4	―	―
Kanzawa et al. [22] (2018)	106.8	67.5	―
Jenni et al. [42] (2020)	104.9	78.4	91.7
Wang et al. [39] (2019)	83.0	57.5	―
Rhodin et al. [24] (2018)	80.1	65.1	―
Wandt et al. [28] (2019)	89.9	65.1	―
Wandt et al. [41] (2021)	81.9	53.0	―
Kocabas et al. [29] (2019)	76.6	67.5	75.2
Lqbal et al. [40] (2020)	69.1	55.9	66.3
Ours (fully supervised)	51.8	41.9	49.4
Ours (self-supervised)	74.6	63.8	73.2

**Table 3 sensors-23-03057-t003:** Pre-training results for the residual network with different depths.

Backbone	PCK (%)
ResNet18	84.7
ResNet34	86.3
ResNet50	88.3
ResNet101	88.9

**Table 4 sensors-23-03057-t004:** Results of ablation experiments.

Transfer Learning	Synthesis Occlusion	WASP	CBAM	MPJPE (↓)	P_MPJPE (↓)	N_MPJPE (↓)
―	―	―	―	102.7	89.5	98.0
√	―	―	―	83.7	71.7	82.8
√	√	―	―	76.1	65.8	74.5
√	―	√	―	83.3	71.2	82.3
√	―	―	√	82.8	73.2	81.8
√	√	√	―	76.0	65.0	74.4
√	√	―	√	74.8	64.1	72.9
√	√	√	√	74.6	63.8	73.2

## Data Availability

http://vision.imar.ro/human3.6m (accessed on 10 February 2022).

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
