# Peer review of "Pose ResNet: 3D Human Pose Estimation Based on Self-Supervision"

_sensors, 2023, doi:10.3390/s23063057_

Round 1
Reviewer 1 Report
The paper has a proper structure containing the introductory section containing the problem statement and a short overview of the existing state of art, the methods applied, the experiments conducted, and conclusions.
It would be appreciated revealing the drawbacks of the known methods and precise what parameters the authors want to improve.
The presentation of the proposed network structure could be, in my opinion, more clear. Provide any reason why did you decide applying ResNet50 structure as a backbone of your structure. Figures 1 and 2 contain the same information; the first one is too general, whereas the second contains too many details. It would be good merging them and point out where the output is and what is the output information. The resolution of the input images is unknown. Are there any premises or indicators in defining Mc(F) (Eq.1) and Ms(F) (Eq.2), PCK(Eq.7)? How "V" is defined (Eq.3)? Upon what reason the threshold value in Eq. 7 was set to 0.5?. In Eq. 8, MJPE (or MJPE(f,s) if you want to emphasize the MJPE dependence) should be on the left side of the equation rather than EMJPE(f,s). Moreover, the ending "2" should not be written as a subscript for the expression under the sum is squared. What "||.||" mean in Eq.8?
Provide any reasons using the percentage of correct keypoints (PCK) as a quality pointer. It would be okay under the assumption that the distance between the predicted coordinate value and ground truth is nearly the same for all cases.
There is no information about the network learning process (a learning plot would be greatly appreciated – loss vs. epoch). Without it, it's hard to assess the tested model quality.
Try to improve the style of writing. Avoid statements expressing the superiority of one solution over another without providing the hard facts or the appropriate reference(s) like, e.g., "The generalization ability of a model trained on the dataset captured by a MOCAP system in a laboratory environment is poor, and the network is unable to estimate special poses that do not appear in the dataset."
Pay more attention to the presentation layer of the paper. Mixing the font styles when writing the variables in the equations and the text is unacceptable.
Reviewer 2 Report
Please cite the dataset used in figure 1
Figure 7 should be cited
Explanation of figure 9 is to be done correctly
Reviewer 3 Report
In this paper, the authors propose a self-supervised 3D pose estimation model named Pose ResNet, which is based on the ResNet50-CBAM and WASP modules, and leverages epipolar geometry to construct 3D labels for self-supervised training. Additionally, transfer learning and data augmentation strategies are employed to enhance the performance of the model.
The idea is not bad and the experimental results are fine. However, there are some problems need to be improved. Specifically:
1) The results of the ablation experiments do not seem to prove that the WASP module improves the performance of the model, and it seems that the experiment 'Transfer learning+WASP' is missing.
2) Tables are hard to read in the first place. E.g., when comparing the results of different models in Table 1, I suggest highlighting the best of the evaluation indicators.
3) Symbols are not standardized, e.g., in Figure 5, '×' should be used instead of '*' when referring to a 3 × 3 kernel.
4) Some related works are missed to compare and analyze, e.g.:
[R1]Weakly-supervised 3D human pose estimation with cross-view U-shaped graph convolutional network. IEEE Transactions on Multimedia 2022.
[R2]PoseTriplet: co-evolving 3D human pose estimation, imitation, and hallucination under self-supervision. CVPR 2022.
[R3]Consistent 3D Hand Reconstruction in Video via self-supervised Learning. IEEE Transactions on Pattern Analysis and Machine Intelligence, 2023.
[4]Joint-bone Fusion Graph Convolutional Network for Semi-supervised Skeleton Action Recognition. IEEE Transactions on Multimedia, 2023. DOI: 10.1109/TMM.2022.3168137.
Round 2
Reviewer 1 Report
I'm fully satisfied with the corrections made in the updated version of the paper.
Author Response
Thank you
Reviewer 3 Report
The authors have addressed most of my concerns. However, there are still two problems:
1) The writing needs to be polish.
2) Some related references are missed to cited, e,g.:
[R1]Z. Tu, Z. Huang, Y. Chen, D. Kang, L. Bao, B. Yang, J. Yuan. Consistent 3D Hand Reconstruction in Video via self-supervised Learning. IEEE Transactions on Pattern Analysis and Machine Intelligence (TPAMI), 2023. DOI: 10.1109/TPAMI.2023.3247907.
